# Preclinical In Vivo Modeling of Pediatric Sarcoma—Promises and Limitations

**DOI:** 10.3390/jcm10081578

**Published:** 2021-04-08

**Authors:** Roland Imle, Felix K. F. Kommoss, Ana Banito

**Affiliations:** 1Hopp Children’s Cancer Center, Heidelberg (KiTZ), 69120 Heidelberg, Germany; r.imle@kitz-heidelberg.de (R.I.); felix.kommoss@kitz-heidelberg.de (F.K.F.K.); 2Pediatric Soft Tissue Sarcoma Research Group, German Cancer Research Center (DKFZ), 69120 Heidelberg, Germany; 3Faculty of Biosciences, Heidelberg University, 69120 Heidelberg, Germany; 4Department of General, Visceral and Transplantation Surgery, Division of Pediatric Surgery, University Hospital Heidelberg, 69120 Heidelberg, Germany; 5Institute of Pathology, University of Heidelberg, 69120 Heidelberg, Germany

**Keywords:** pediatric sarcoma, animal models, GEMMs, PDXs, preclinical testing

## Abstract

Pediatric sarcomas are an extremely heterogeneous group of genetically distinct diseases. Despite the increasing knowledge on their molecular makeup in recent years, true therapeutic advancements are largely lacking and prognosis often remains dim, particularly for relapsed and metastasized patients. Since this is largely due to the lack of suitable model systems as a prerequisite to develop and assess novel therapeutics, we here review the available approaches to model sarcoma in vivo. We focused on genetically engineered and patient-derived mouse models, compared strengths and weaknesses, and finally explored possibilities and limitations to utilize these models to advance both biological understanding as well as clinical diagnosis and therapy.

## 1. Introduction

Sarcomas are mesenchymal malignancies accounting for about 15% of cancers in children and adolescents, making them the third most common group of childhood cancers, following blood malignancies and brain tumors [1]. While the last decades have seen vast improvements in pediatric cancer care with overall improved prognosis, this does not hold true for sarcomas, which are often prone to metastasis and relapse, typically accompanied by dismal prognosis [2]. Research efforts to improve this situation are complicated by the extremely diverse intrinsic nature of pediatric sarcoma with more than 60 genetically distinct entities [3]. While some pediatric sarcoma types may show widespread genomic instability (e.g., osteosarcoma (OS)), many are genetically rather simple, characterized by pathognomonic fusion oncogenes (e.g., SSX-SS18 in Synovial Sarcoma (SySa)) [4,5,6]. Ongoing molecular profiling efforts will likely lead to further sub-classification as we learn more about specific genetic and epigenetic alterations and underlying biology [7]. Figure 1 provides a snapshot of the most common pediatric sarcoma types according to the recently (2020) updated World Health Organization (WHO) classification of soft tissue and bone tumors (Figure 1a) [8,9], as well as an unbiased molecular clustering of the most common tumor entities based on DNA methylation data (Figure 1b) [10]. The development of novel therapeutic agents heavily relies on preclinical testing in disease specific models. Given the rarity of each individual pediatric sarcoma subtype, appropriate model systems are naturally scarce, and the selection of suitable models is challenging. This represents a major problem for pediatric cancer research and has significantly contributed to the lack of meaningful therapeutic improvements in pediatric sarcomas [11]. Therefore, in this review, we aim to outline the pros and cons of major in vivo modeling approaches applicable for pediatric sarcoma. We review existing models applied for specific sarcoma entities and discuss relevant points to consider for meaningful future model utilization. Due to the vast array of known malignancies, the scope of this review is limited to 18 clinically particularly relevant sarcoma entities.

## 2. In Vivo Modeling Approaches Applicable for Pediatric Sarcoma

Analogous to other solid tumors, four main general approaches of in vivo cancer modeling can be distinguished for pediatric sarcoma (Figure 2) [12,13]. Two of these entail the engraftment of human cancer tissue into immunocompromised mice—cell-line-derived xenograft models (CDXs) and patient-derived xenograft models (PDXs)—while the other two induce de novo tumorigenesis in immunocompetent wild type mice—environmentally-induced models (EIMMs) and genetically-engineered mouse models (GEMMs). Figure 3 highlights the major advantages and disadvantages of these approaches. Independent of the specific approach, one should consider that different mouse strains, much like humans, possess an inherent and strain-specific risk of spontaneously developing different cancer entities over their lifespans [14,15]. While these can, in some cases, also serve as useful models of human cancer, they should by no means be mistaken for specifically engrafted human or induced murine tumor tissue [16].

CDXs are the most commonly used, but least representative model when aiming to recapitulate the original disease. EIMMs are extremely powerful, but since pediatric sarcomas are usually not driven by environmental factors, they are not as relevant for childhood sarcoma. PDXs and GEMMs however, are highly representative of their human counterpart, therapeutically predictive and complementary to each other in nature.

### 2.1. Genetically-Engineered Mouse Models (GEMMs)

GEM models follow the principle of activating or inactivating specific cancer-associated genes in immunocompetent mice to study their impact on tumorigenesis in vivo. While early GEMM-approaches were restricted to costly embryonic stem cell alterations and extensive crossing of chimeric offspring to yield hetero- and homozygous animals for a given gene of interest [17], later work developed transgenic mouse lines expressing Cre under a variety of tissue-specific promoters, allowing conditional gene activation (typically LoxP-Stop-LoxP-Promoter-gene) or inactivation (typically LoxP-gene-LoxP) via crossing of mice [18,19]. However, germ-line models conveying constitutive gene regulation often exhibit embryonic lethality and developmental defects due to the important developmental role of many oncogenes and tumor suppressors genes [20,21,22,23,24]. This problem was overcome by the introduction of tamoxifen-inducible Cre lines, which express Cre in a Tamoxifen-inducible fashion from the endogenous *Rosa26*- or other more tissue-specific promoters, allowing time- and space-dependent gene regulation pre- and postnatally [25]. Profound advances in model technology entail the application of somatic gene editing approaches via lenti- and adenovirus-delivery or in vivo electroporation [26,27] and their constant optimization for in vivo use [28]. Importantly, Huang et al. could show that both CreLoxP-mediated recombination as well as somatic clustered regularly interspaced short palindromic (CRISPR)/Cas9-mediated gene editing do not lead to differences in the molecular makeup or biological behavior of induced tumors [29]. A holistic review on GEMMs in cancer research is provided by Kersten et al. [30].

### 2.2. Patient-Derived Xenografts (PDXs)

In PDX models, primary patient tumor material is transplanted into immunocompromised mice. Although PDXs cannot recapitulate the early steps of tumorigenesis in an intact immune microenvironment like GEMMs, their standout strength is the recapitulation of heterogeneity of their human counterparts with each PDX reflecting an individual patient [31]. This makes them a particularly valuable resource enabling the correlation of intensive molecular profiling with preclinical therapy response data to direct rational clinical trial design and select the correct patient cohorts for targeted treatments [32,33,34]. While most PDXs are established by subcutaneous (s.c.) rather than orthotopic engraftment due to easier handling and tumor surveillance, the orthotopic microenvironment may be beneficial for tumor take rates and for preserving tumor biology, as comprehensively analyzed by Stewart et al. [35]. From a technical point of view, one has to consider that tumor take rates vary greatly between about 20% and 60%, with mostly of successful engrafted tumors corresponding to aggressive/relapsed/metastasized cases. Further, stable PDX-establishment typically requires four passages in vivo, making the entire process from first engraftment to final establishment lengthy and cost-intensive, but nonetheless worthy [34,36,37,38]. Maybe most importantly, both GEMM and PDX models are regarded highly predictive for clinical therapy response [30,33,38,39].

## 3. Established In Vivo Models of Pediatric Sarcoma

To find relevant PDX repositories entailing pediatric sarcomas (Figure 4) and to identify established GEM-modeling approaches for 18 relevant pediatric sarcoma entities (Appendix A), a systematic literature search was performed. Additionally, the most up-to-date previous sarcoma mouse model reviews by Dodd et al. (2010) [19], Post (2012) [39], Seitz et al. (2012) [40], Brossier et al. (2012, further focused on Malignant Peripheral Nerve Sheath Tumor (MPNST) and related entities) [41], Jacques et al. (2018, further focused on bone sarcoma) [42], and O’Brien [43], Zanola et al. [44], and Yohe et al. [45] (2012/2012/2019, further focused on rhabdomyosarcoma (RMS)), were carefully considered.

### 3.1. Existing Cell-Line-Derived Xenograft Models (CDXs) for Pediatric Sarcoma

As extensively review by Gengenbacher and Singhal et al., the current mainstay of in vivo cancer modeling are engraftments of long-established cell lines from often cultured over many passages, using high amounts of fetal calf serum (FCS) in vitro [13]. This also holds true for sarcoma with over 70% of articles entailing sarcoma mouse models in high-ranking journals in 2016 applied CDX models both for basic and translational research while utilization of PDXs and GEMMs was under 10% [13]. This is often due to broad availability and experience as well as ease of use. Common “work horses” of sarcoma cell lines also used for in vivo engraftment are Rh30 (alveolar rhabdomyosarcoma—aRMS), A204 (embryonal rhabdomyosarcoma—eRMS), RD (RT), HS-SY-II (SySa), TC71 (Ewing Sarcoma—EwS) and KHOS (osteosarcoma—OS) among many others. Many of these models, however, do not faithfully represent the original tumor and are highly adapted to two-dimensional (2D) culture conditions. TC71, for example, one of the most commonly used EwS cell lines, harbors a BRAF mutation, which is rarely found in EwS patients and can influence therapy response [46,47]. Hinson et al. provide a holistic review on commonly used RMS cell lines and necessary precautions to consider, which can also be applied to other sarcoma entities [48].

### 3.2. Existing Patient-Derived Xenograft Models (PDXs) for Pediatric Sarcoma

PDX models of various entities, including sarcoma, are undisputedly highly valuable tools towards a deeper understanding of cancer biology and treatment advances [33]. The current key challenge, however, is their availability with many fragmented efforts to collect and transplant patient samples into mice, scattered over the world [49]. For sarcoma, this is particularly challenging due to the rarity of specific sub-types and the logistic challenges to obtain these samples, particularly in countries with decentralized pediatric cancer care [32,35,38,50]. Figure 4 provides a list of PDX repositories entailing pediatric sarcoma. The Memorial Sloan Kettering Cancer Center’s Department of Pediatrics will also soon be releasing their collection of pediatric PDX models including several sarcoma entities such as desmoplastic small round cell tumor (DSRCT) [51].

Some models can also be retrieved amidst larger repositories, mostly entailing models for common adulthood cancers. While many of these repositories have large amounts of models and accompanying data (e.g., https://www.europdx.eu/, last accessed 21 April 2021, with 1500+ models) and try to bridge the efforts of academic institutions and contract research organizations (e.g., https://repositive.io/, last accessed 21 April 2021, with 8000+ models), PDXs of pediatric malignancies remain largely underrepresented or not present at all. Furthermore, since many do not allow filtering available models by age group, it appears that they are often not poised to encourage deposition of pediatric PDXs.

Most repositories also supply comprehensive data sets on the molecular characterization of PDXs. This is particularly important to increase their value as preclinical testing tools. To this end, the recently proposed so-called “PDX models minimal information standard” (PDX-MI), defining a basic standard of PDX model description, could be of great value to help researchers to pick the right models for their respective question [52].

### 3.3. Existing Environmentally-Induced Mouse Models (EIMMs) for Pediatric Sarcoma

Environmentally-induced sarcoma models are mostly relevant for adulthood sarcoma since sarcoma in children and adolescents is typically the result of distinct genetic events rather than accumulation of genetic alterations as a result of environmental factors. Zanola et al. and Kemp et al. provide reviews entailing several EIMM systems, relevant for adulthood cancers [44,53]. Nonetheless, intramuscular (i.m.) injection of both cardiotoxin (CTX) and barium chloride to induce muscle damage and subsequent muscle regeneration with a more activated state of satellite cells (major stem cell pool for muscle regeneration), have been successfully used to induce a regenerative environment with increased susceptibility towards undifferentiated pleomorphic sarcoma (UPS) and RMS in several GEM modeling approaches (see GEMMs section) [54,55,56,57]. Interestingly, genetic models of muscular dystrophy also seem to provide a micromilieu and cellular state, which clearly facilitates sarcomagenesis with a remarkable specificity towards eRMS, which even correlates with severity of muscle dystrophy (see GEMM section) [54,58,59,60].

Radiotherapy and chemotherapy can also be regarded as external cues of mutagenic nature that children being treated for cancer frequently face on an everyday basis [61]. Since close to 10% of pediatric cancer is likely based on predisposing germ line variants, many of which can increase the susceptibility to mutagenic cues (e.g., radiotherapy in neurofibromatosis), the relevance of this could be largely underestimated [62]. To this end, Lee et al. presented a very informative comparison of murine sarcoma induction by either radiation or local injection of the mutagen 3-methylcholanthrene (MCA) in a wild type or *Tp53*-null background as well as a genetic model of *Kras* overexpression and *Tp53* knockout. These comparisons revealed distinct mutagenic patterns and different levels of genomic stability, depending on the causative event [63].

### 3.4. Existing Genetically-Engineered Mouse Models (GEMMs) of Pediatric Sarcoma

Both the cell of origin, which is often not entirely known for many sarcomas, and mutational profile likely determine sarcoma biology and appearance [18]. While many sarcomas are determined by pathognomonic driver oncogenes, such as *PAX3-FOXO1* in aRMS, UPS, and eRMS are not as clearly genetically defined. Additionally, the same genetic alterations (e.g., oncogenic *RAS* mutation plus *TP53* inactivation) can lead to both UPS, pleomorphic RMS and eRMS, and can be seen as a disease spectrum of varying divergence in cell of origin and mutational profile [18]. Details on existing GEMMs of the 18 focus entities researched for the scope of this review is provided in Appendix A.

#### 3.4.1. Undifferentiated Pleomorphic Sarcoma (UPS)

UPS refers to aggressive undifferentiated soft tissue and bone sarcomas, which lack an identifiable line of differentiation. While UPS are more common in adults, they may also occur in children [64]. The early work of genetic cancer modeling in mice focused on the tumor spectrum of mice deficient for major tumor suppressors such as *Tp53*, including specific mutations mimicking Li–Fraumeni syndrome [65]. While this does not induce specific tumor entities, but rather a plethora of different cancers with increased penetrance compared to unaffected mice, the most commonly occurring neoplasms are lymphomas and sarcomas. Sarcomas typically possess UPS morphology, more rarely also OS-, RMS-, and angiosarcoma appearance [17]. Sarcoma penetrance varied from about 10 to 50% when mice were surveilled over their entire lifespan [66,67]. If additional tumor suppressors, such as *Pten*, are knocked out, the efficiency increases dramatically (100% penetrance/10 weeks median latency) [68]. Furthermore, oncogenic Ras could be identified as one of the strongest oncogenes, requiring co-occurring tumor suppressor silencing (e.g., *Tp53* or *Rb1*) to avoid apoptosis and senescence [27,69,70]. While the introduction of *TP53* hotspot mutations are even more efficient in tumorigenesis than *Tp53* loss and lead to spontaneous metastasis in about 13% of cases [70], mutant Ras can also cooperate with *Cdkn2a* inactivation to induce UPS with similar efficiency [26]. Applying *Myf7*-and *MyoD-CreER* lines, Blum et al. identified Myf7-positive muscle progenitors as a cell of origin for both UPS (62%) and RMS (38%, 63% of which were graded eRMS), while MyoD-positive progenitors only led to UPS (70% of which showed myogenic features) [55]. This is believed to correlate with activation status of satellite cells as the muscle regeneration stem cell pool, which could also be induced by i.m. injection of cardiotoxin to induce muscle damage and regeneration.

#### 3.4.2. Embryonal/Fusion-Negative Rhabdomyosarcoma (eRMS) and Pleomorphic RMS

Embryonal rhabdomyosarcoma is the most common RMS subtype and genetically more diverse than aRMS and other sarcomas. This is reflected by the plethora of different eRMS models induced by different oncogenes and tumor suppressors over the last 25 years [71]:•Sonic Hedgehog signaling: interestingly, one of the first identified RMS GEMMs with embryonal morphology was incidentally found in a *Ptch*-inactivated mouse model of Gorlin syndrome, an autosomal dominant syndrome predisposing towards basal cell carcinoma, medulloblastoma, and RMS. This model developed eRMS with an incidence of 9% and 1% in CD-1 and C57BL/6 mice respectively, also highlighting the relevance of mouse strain differences for studying tumorigenesis [72]. Since then, several papers were built upon this work by dissecting the major components of Sonic Hedgehog (SHH) signaling and their influence on tumorigenesis. A tamoxifen-inducible model from Mao et al. showed that expression of a constitutively active form of the cellular signal transducer Smoothened (Smo), called SmoM2, can drive eRMS on a *Ptch^−/+^*-background with 100% penetrance within five weeks [73]. Releasing Sufu (Suppressor of fused) inhibition on the Gli effector proteins (*Sufu*^−/+^) can also drive eRMS on a *Tp53*^−/−^ background while *Sufu*^−/−^ is embryonically lethal [23]. Thus, as further worked out by the inducible model of *Ptch*-inactivation by Zibat et al., *Sufu* mutations appear more efficient in sarcomagenesis than *Ptch* mutations [74]. Interestingly, the remaining wild type allele of *Ptch*^−/+^-mice was also silenced in the course of eRMS-tumorigenesis. Hatley et al. showed that SmoM2 can even induce eRMS when expressed in the adipocytic lineage (*aP2*-Cre) with about 80% or up to 100% penetrance when cooperating with loss of *Cdkn2a* [75]. Rubin et al. utilized various Cre-drivers to investigate conditional *Ptch*-and *Rb1*-inactivation on a *Tp53*-null background and found a tumor spectrum of eRMS, UPS, and partly OS with satellite cells predisposed towards UPS, and maturing myoblasts towards eRMS development [18]. Finally, Fleming et al. recently showed that expression of *Gli2A*, a constitutively active form of SHH effector protein Gli2, via *PCP2*-Cre leads to small round cell tumors with Ewing-like features with nearly 100% penetrance and a median latency of about 8 weeks [24]. Pairing with SmoM2-expressing mice was embryonically lethal.•eRMS in models of muscle dystrophy and regeneration: Chamberlain et al. made the incidental observation that about 6% of muscular dystrophy X-linked (MDX) mice, which model Duchenne muscular dystrophy by harboring a spontaneous point mutation in exon 23 of the dystrophin gene, develop eRMS late in life [58]. Fernandez et al. validated this finding for MDX mice (9% eRMS penetrance late in life) and further found that mice deficient for of Alpha-Sarcoglycan (*Sgca*^−/−^), mutated in limb girdle muscular dystrophy (LGMD) can also lead to eRMS occurrence late in life (4% penetrance). Tumors exhibit *Mdm2* and P53 amplification with cancer-associated P53 missense mutations. Camboni et al. bred MDX mice to homozygous or heterozygous *Tp53* knockout mice and found 60%/26 weeks and 90%/17 weeks of penetrance and median latency, respectively [54]. Efficiency of tumorigenesis could further be increased by inducing muscle damage and regeneration by intramuscular CTX injection (100% penetrance/13 weeks median latency). This led to the idea that the regenerative cellular/microenvironmental state induced by muscle repair sensitizes towards sarcomagenesis. Recently, Boscolo et al. applied a more severe model of Duchenne muscular dystrophy (MDX/MtR mice) and combined it with injection of Barium chloride to induce muscle damage and regeneration [60]. Strikingly, muscle stem cells acquired an RMS-like gene signature before transformation, leading to very efficient eRMS-tumorigenesis (100%/17 weeks median latency). Van Mater et al. applied a dual recombinase system for a UPS model driven by *KRAS^G12D^* and/or *Tp53* inactivation and found that muscle injury can to some degree substitute for *KRAS^G12D^* and leads to UPS with chromosomal gains encompassing *Yap1* and *Met* [76]. Collectively, the work of Rubin, Boscolo, Tremblay, Van Mater and Blum et al. clearly show a connection between muscle regeneration, activation status of satellite cells and the susceptibility of sarcomagenesis towards eRMS and UPS [18,55,56,60,76].•Hippo signaling: Tremblay et al. found that this paradigm also holds true for eRMS driven by a constitutively active mutant of *YAP1*(S127A). Strikingly, YAP1 (Yes-associated protein 1) hyperactivity only induced sarcoma in activated satellite cells after i.m. injection of CTX or barium chloride, but not in quiescence [56]. Slemmons et al. further found cooperation between activated YAP1 and oncogenic Ras in eRMS [57].•Her2/neu signaling: Nanni et al. and Ianzano et al. show that Her2/neu (human epidermal growth factor receptor 2) can drive urogenital eRMS on a *Tp53*^−/+^ background with high efficiency (100%/14–17 weeks median latency) with a remarkable gender specificity for males [77,78].•Hepatocyte-growth-factor-receptor (Hgfr)/c-Met-signaling: Takayama et al. showed that overexpression of hepatocyte growth factor/scatter factor (*HGF/HF*) leads to induction of various malignancies, including RMS (7% penetrance) via autocrine c-Met signaling [79]. Sharp et al., further found that *Cdkn2a* inactivation strongly increases the efficiency of eRMS induction upon c-Met-activation (90% penetrance/14 weeks median latency) [80].•P53 cooperation: as outlined above, various signaling pathways cooperate with *Tp53*-inactivation for sarcomagenesis. Comiskey et al. recently developed a model highlighting this relationship, featuring *MDM2-ALT1* (splice variant 1 of murine double minute 2), which is frequently expressed in eRMS (70%) and aRMS (85%) [81]. When expressed via *Sox2*-Cre it promotes eRMS-induction in a *Tp53*^−/+^ (100% penetrance/20 weeks median latency, 50% of tumors are eRMS), but not *Tp53*^−/−^-background (100% penetrance/27 weeks median latency, mostly lymphoma, no eRMS. Further models include et a double knockout model of *Fos/Tp53*, developing eRMS in facial and orbital regions by Fleischmann et al. [82] and a cardiac model of RMS (not further specified) by Köbbert et al., who inactivated both *Tp53* and *Rb1* via microinjection of embryos with SV40-Tumor antigen (TAg) under the *Sm22alpha* promoter (active in both smooth muscle and embryonic cardiac muscle) [83].

#### 3.4.3. Alveolar/Fusion-Positive Rhabdomyosarcoma (aRMS)

The aRMS constitute the second most common RMS subtype and usually occurs in adolescents and young adults (peak incidence at 10–25 years of age) [84]. aRMS exhibit skeletal muscle differentiation and specific molecular alterations (either a *PAX3-FOXO1* or a *PAX7-FOXO*1 gene fusion) are detected in the majority of cases [85]. Despite this clear molecular definition, the cell of origin of aRMS is not entirely clear, complicating mouse modeling development [86]. Lagutina et al. and others found that constitutively expressing the *Pax3-Foxo1* fusion from the endogenous *Pax3* locus leads to developmental defects, but not tumorigenesis [87]. Heterozygous and chimeric mice showed developmental muscle defects and died perinatally from cardiac/respiratory failure. Interestingly, *Pax3-Foxo1* expression from exogenous *PGK*-, *MyoD*-and rat-*beta-actin* promoter did not yield any phenotype.

Keller et al. later validated the developmental role of *Pax3-Foxo1* upon embryonic expression and further applied a conditional model using *Pax7*-Cre to knock-in *Pax3-Foxo1* into the endogenous *Pax3* locus [88]. This expression, starting in terminally differentiating muscle cells, gave rise to aRMS, although with extremely low penetrance (1 of 228 mice, about a year after birth) [89]. Additionally, inducing haploinsufficiency for Pax3 by a second conditional allele did not accelerate tumorigenesis. Inactivation of *Cdkn2a* and even more so of *Tp53* however, increased efficiency markedly to about 30–40% when carried out on both alleles [89]. This model was further characterized molecularly by Nishijo et al. who found a preserved gene expression signature between human and murine aRMS and observed that spontaneous metastasis, albeit occurring at very low frequency, was selected for high expression of the *Pax3-Foxo1*-fusion [90].

A follow-up study by Abraham et al. from the Keller laboratory further utilized this conditional *Pax3-Foxo1* model, using four different Cre lines [91]: *M*Cre (Pre-and postnatal hypaxial lineage of *Pax3* that includes postnatal satellite cells), *Myf5*-Cre (Pre- and postnatal lineage of Myf5 that includes quiescent and activated satellite cells and early myoblasts), *Myf6*-CreER (Pre-and postnatal lineage of Myf6 that includes maturing myoblasts) and *Pax7*-CreER (Postnatal lineage of Pax7 that includes quiescent and activated satellite cells) [18]. While tumors of *M*Cre (40% penetrance/median 29 weeks) and *Myf6*-CreER (100% penetrance/median 15 weeks) showed typical aRMS morphology, *Pax7*-CreER had spindle/pleomorphic appearance, reminiscent of fusion-negative RMS and prolonged tumor-free-survival (65% penetrance/median 48 weeks) [91]. Strikingly, reporter gene- and fusion gene expression was also lower in *Pax7*-CreER tumors, indicating lower transcription from the *Pax3* locus in these mice and conveying divergent therapy response to histone deacetylase (HDAC) inhibitor entinostat. *Myf5*-Cre mice were embryonically lethal with only one mouse surviving and developing aRMS amidst increased anaplasia after 10.5 weeks:•Hippo signaling: Oristian et al. further applied the *Myf6*-Cre-driven conditional *Pax3-Foxo1*/*Cdkn2a*^Flox/Flox^/*Pax3*^Pf/Pf^ model of aRMS and added *Stk3*^Flox/Flox^ and Stk4^Flox/Flox^ to activate Hippo signaling [92]. This resulted in increased tumorigenesis (88%/median 16 weeks) vs. 27%/median 26 weeks) and an increased number of tumors per animal, highlighting the role of Hippo signaling in aRMS.

Unfortunately, so far, none of the described RMS models could shed light on the age discrepancy between eRMS and aRMS patients.

#### 3.4.4. Spindle Cell/Sclerosing RMS with MYOD1 Hotspot Mutation

*MYOD1*-mutant RMS represents a distinct subtype of spindle cell and sclerosing RMS. While the recurrent hotspot mutation of this biologically distinct RMS is known and appears to result in a particularly aggressive clinical course, no GEM modeling attempts could be identified in the literature to date [93]. Only a single extensively characterized patient-derived cell line model was identified [94].

#### 3.4.5. Osteosarcoma (OS)

Osteosarcoma, a bone sarcoma characterized by a complex karyotype, can occur in children/young adults or later in life (about 60 years of age). OS is associated with genetic predisposition, in particular to Li-Fraumeni and retinoblastoma syndromes and most OS exhibit mutations/deletions of *TP53* and/or *RB* [95,96,97]. Accordingly, several murine models relying of *Tp53* inactivation have been applied to study osteosarcoma. Despite the fact that *Tp53* null mice are prone to several malignances, it has been reported that 4% develop osteosarcomas (OS showing longer latency than other malignancies) with a higher frequency of OS (25%) in *Tp53* heterozygous mice [17,65]. Consistent with a key role for p53 in osteosarcoma, mice harboring the *p53R172H* gain of function mutant knock-in develop osteosarcoma able to metastasize to other organs [67]. On the other hand, mice heterozygous for *Rb* deletions are not predisposed to OS, while mice homozygously deleted for *Rb* die at birth [98,99]. The development of several cell lineage-specific Cre expressing lines, allowed to develop many additional and improved GEMMs where *Tp53* and/or *Rb* are inactivated in the mesenchymal/osteogenic linage; therefore, more faithfully resembling the human disease. Inactivation of *Tp53* alone or together with *Rb* in *Prx-1* positive cells (mesenchymal/skeletal progenitors) *Osx*, *Col1A1*, or *Og2* positive cells (pre-osteoclasts and osteoclasts) generates OS with high penetrance often leading to metastatic disease [100,101,102,103,104,105,106]. For a detailed summary of genetically engineered mouse models for OS, see Appendix A, and recent reviews provided by Guijarro et al. [107] and Uluçkan et al. [108].

#### 3.4.6. Ewing Sarcoma (EwS)

EwS is a small round cell sarcoma most commonly arising in the bone of children and young adults (most cases < 20 years of age). Extraskeletal manifestation of EwS occurs in about 12–20% of affected patients [109]. Ewing sarcoma’s pathognomonic driving oncogene *EWSR1-ETS* (typically *FLI1*) is functioning as a neo-transcription factor as well as an epigenetic regulator by inducing de novo enhancers at GGAA microsatellites [109]. Unfortunately, all 16 alternative attempts in 6 independent laboratories to create a transgenic Ewing sarcoma mouse model failed to date—comprehensively presented in a joined manuscript by Minas et al., 2017 [22]. Most attempts did not lead to any tumorigenesis at all despite using various tissue-specific promoters to target the potential cells of origin in EwS in different stages of development: Mesenchymal stem cells (MSCs), neural crest stem cells and embryonic osteochondrogenic progenitors [110]. Developmental *EWSR1-FLI1* expression typically led to embryonic lethality while conditional expression in later stages led to various developmental defects, such as muscle degeneration. A modeling approach with successful tumorigenesis applied *EWSR1-FLI1* transduction into bone marrow derived MSCs followed by intravenous (i.v.) injection into sub-lethally irradiated mice. However, it did not result in EwS, but fibrosarcoma located in the lung.

Possible explanations for these marked difficulties in model generation could lie in promoter leakiness, the lack of potential co-factors, but also in distinct biological differences between mice and humans, such as divergent splice acceptor sites, low CD99 homology, and unequal GGAA microsatellite architecture, which is not well conserved between species [111]. Particularly, the important epigenetic regulator function of EWSR1-FLI1 depends on an appropriate number of chromatin-accessible GGAA microsatellites in proximity to relevant genes to allow transformation without inducing apoptosis and might not be appropriate in mice. The fact that in vitro transformation of murine cells, such as osteochondrogenic progenitors, is possible and to some extent resembles human EwS, suggests that creating an endogenous EwS GEMM could be conceivable [112,113].

#### 3.4.7. Synovial Sarcoma (SySa)

SySa represents a spindle cell sarcoma with variable epithelial differentiation, harboring a pathognomonic *SS18-SSX1/2/4* gene fusion. Haldar et al. showed that when *SS18-SSX2* expression is induced in Myf5-expressing myoblasts, 100% of mice develop synovial sarcoma-like tumors [114]. Importantly, the induction of *SS18-SSX2* expression through *Hprt*-Cre, *Pax3*-Cre, or *Pax7*-Cre resulted in embryonic lethality, while *SS18-SSX2* activation in *Myf6*-expressing myocytes or myofibers resulted in myopathy but no tumors. Therefore, this fusion is able to induce tumorigenesis in mice when expressed at the right time, in the right cell population (permissive cellular background) [114,115,116]. In the *Myf5*-Cre linage, tumors formed with 100% penetrance, presented both biphasic and monophasic histology and expressed a gene signature that partially overlapped with that of human synovial sarcoma [114]. Locally induced expression of *SS18-SSX1* or *SS18-SSX2* using TATCre injection also yields tumors, however with longer latency than Myf5-Cre mice. Exome sequencing identified no recurrent secondary mutations in tumors of either genotype (*SS18-SSX1/2*) further highlighting the idea that the fusion alone is able to drive the disease in a specific permissive background [117]. Using this localized induced model Barrott et al. showed that *Pten* silencing dramatically accelerated and enhances sarcomagenesis without compromising synovial sarcoma characteristics and additionally leading to spontaneous lung metastasis [118]. The same laboratory further showed that co-expression of a stabilized form of β-catenin greatly enhances synovial sarcomagenesis by enabling a stem-cell phenotype in synovial sarcoma cells, blocking epithelial differentiation and driving invasion [119].

#### 3.4.8. Malignant Peripheral Nerve Sheath Tumor (MPNST)

Sarcomas rarely follow the benign-to-malignant multistep progression course, prototypical for many of the “big killers” in adult oncology (e.g., colorectal carcinoma). MPNST, however, can partly be regarded an exception to this rule as it often develops from plexiform or dermal neurofibromas, which represent benign lesions with homo- or heterozygous deletion of the tumor suppressor neurofibromin (*NF1*), acting inhibitory to Ras-signaling through its GTPase activity [120]. This can mostly be attributed to the fact that a large proportion of MPNSTs arises in neurofibromatosis patients, an autosomal dominant disease caused by inactivating *NF1*-mutations, but also to the fact that the unusually large *NF1* gene is among the most frequently mutated genes of the human genome [121]. Additional genetic hits, such as *TP53*-or *Cdkn2a*-inactivation induce malignant progression of neurofibromas. In mice, while *Nf1*-plus *Tp53* deletion in Schwann cells leads to MPNST, inactivation in more mesenchymal progenitors or muscle cells leads to *Nf1*-inactivated RMS or UPS [122]. NF1 and P53 are also exemplary of the developmental importance of many tumor suppressor genes and oncogenes. The first *Nf1/Tp53* knockout models, described in 1999 by both Cichowski et al. and Vogel et al., observed embryonic lethality upon inactivation of these genes in embryonic development [20,21]. Later conditional knockout models using Cre lines specific to different tissue-specific promoters established the Schwann cell lineage as the cellular origin of MPNSTs (comprehensively reviewed Brossier et al., 2012) [41]. Since then, Dodd et al. showed that local injection of an adenovirus, delivering Cre into the sciatic nerve of *Tp53*-wild type mice (*Nf1*^Flox/Flox^; *Ink4a/Arf*^Flox/Flox^) also induces MPNST, while intramuscular injection leads to RMS/UPS [122]. Huang et al. validated that MPNST induction can also be obtained via lentiviral delivery of a CRISPR/Cas9 construct, targeting *Nf1* and *Tp53*, when injected into the sciatic nerve of wild type mice [29].

#### 3.4.9. Infantile Fibrosarcoma (IFS)

IFS represents a primitive sarcoma of fibroblastic differentiation in many of which a characteristic *ETV6-NTRK3* fusion is identified [123]. While there have not been any comprehensive GEM models described to date, both *ETV6-NTRK3* as well as the non-canonical fusion gene *EML4-NTRK3* have been shown to be able to transform murine NIH3T3 fibroblasts, which successfully engrafted s.c. in severe combined immunodeficiency disease (SCID) and NOD SCID gamma (NSG) immunocompromised mice to result in tumors with IFS-like histomorphology [124,125].

#### 3.4.10. Malignant Rhabdoid Tumors (MRT)

Malignant rhabdoid tumors (MRT) are aggressive, poorly differentiated pediatric cancers, characterized by the presence of germline/somatic biallelic inactivating mutations or deletions of the *SMARCB1* (*INI1*, *SNF5,* or *BAF47*) gene, which is a component of the SWItch/Sucrose Non-Fermentable (SWI/SNF or BAF) chromatin-remodeling complex. Tumors can arise in the soft-tissue or in the kidney and less commonly in the central nervous system (referred to as atypical teratoid rhabdoid tumor; AT/RT). In mice, *Smarcb1* heterozygous loss predisposes to soft-tissue sarcoma tumors consistent with human MRT but with low penetrance (approximately 12%) [126,127,128]. However, homozygous or heterozygous deletion of *Tp53* (but not of *Cdkn2a* or *Rb1*) in *Smarcb1* heterozygous mice accelerates tumor onset and penetrance [129,130]. Conditional inactivation of Smarcb1 in mice (*Smarcb1*^Flox/Flox^, Mx-Cre plus polyI/polyC treatment) results in rapid cancer susceptibility, with all animals developing tumors at a median age of 11 weeks [131]. These lesions exhibit many features of rhabdoid tumors, such as rhabdoid cells and complete absence of Smarcb1 expression.

#### 3.4.11. Clear Cell Sarcoma of Soft Tissue (CCS)

CCS is an aggressive neoplasm that usually arises in the deep soft tissue of young adults. The genetic hallmark of CCS is t(12;22)(q13;q12) leading to an *EWSR1-ATF1* gene fusion [132]. Straessler et al. published a model for conditional and Tamoxifen-inducible EWSR1-ATF1 expression under the endogenous Rosa26 promoter [133]. Temporal regulation of the fusion gene expression was required due to embryotoxicity of *EWSR1-ATF1*. Conditionally expressing human cDNA of *EWSR1-ATF1* without any accompanying alterations led to highly efficient tumorigenesis of CCS-like tumors with 100% penetrance that resemble human CCS morphologically, immunohistochemically and by genome-wide expression profiling [133].

#### 3.4.12. Alveolar Soft Part Sarcoma (ASPS)

ASPS is a deadly soft tissue malignancy, which consistently demonstrates a t(X;17)(p11.2;q25) translocation that produces the *ASPSCR1-TFE3* fusion gene [134]. Expression of human cDNA in a temporal fashion (*Rosa26*-CreER) leads to ASPS-like tumors that resemble the human disease in terms of histology and expression patterns. Mouse tumors demonstrate angiogenic gene expression and are restricted to the tissue compartments highest in lactate, suggesting a role for lactate in alveolar soft part sarcomagenesis [135].

#### 3.4.13. Rare Sarcomas without Current GEM Models


•Clear cell sarcoma of the kidney (CCSK): CCSK is a rare neoplasm, typically arises in the kidney of infants and young children. CCSK has a dismal prognosis, often showing late relapses [136]. Recently, an internal tandem duplication of exon 15 of BCL-6 corepressor (*BCOR*) was identified as the major oncogenic event in CCSK [137].•Small round blue cell tumor with BCOR alteration (SRBCT-BCOR): SRBCTs represent a heterogenous group of tumors, from which SRBCT-BCOR was only recently defined as a stand-alone entity. SRBCT-BCOR typically harbors *BCOR*-related gene fusions (e.g., *BCOR-CCNB3*) or an internal tandem duplication within Exon 15 of *BCOR* [138,139]. SRBCT-BCOR are rare neoplasms, mostly arising in infants and young children, showing a striking male predominance [140].•Small round blue cell tumor with CIC alteration (SRBCT-CIC): similarly, to SRBCT-BCOR, SRBCT-CIC was recently identified as a distinct subtype of SRBCT [141]. In most cases a *CIC-DUX4* gene fusion is identified [142]. SRBCT-CIC may arise in children and older adults; however, most cases are observed in young adults (25–35 years of age) [143].•Desmoplastic small round cell tumor (DSRCT): DSRCT is a malignant mesenchymal neoplasm, most frequently arising in the abdominal cavity of children and young adults [144]. Typically DSRCT harbors t(11;22)(p13;q12), leading to a *ESWR1-WT1* gene fusion [145].•Mesenchymal chondrosarcoma (MC): MC is a rare neoplasm, typically arising in craniofacial bone and adjacent soft tissues of young adults [146]. The genetic hallmark of MC is a *HEY1-NCOA2* gene fusion [147].•Inflammatory myofibroblastic tumor (IMT): IMT is a myofibroblastic neoplasms arising in various locations, which usually shows a benign clinical course. However, few patients will present with local recurrence and/or distant metastasis [148]. In IMT, gene rearrangements affecting receptor tyrosine kinase genes (most often involving *ALK*) are typically identified [149].


The lack of models for many entities are a consequence of limited knowledge in regards to the tumor cell of origin, high heterogeneity of cellular backgrounds, rapid emergence of new molecular sub-types, and the need for more flexible models that allow the testing of various genetic alterations in different cellular backgrounds.

### 3.5. Non-Murine Animal Models for Pediatric Sarcomas

Mice as modeling organisms enable a great trade-off between appropriate resemblance of their human counterpart, and thereby translatability, as well as decently short generation times and experimental practicality. However, all aforementioned modeling approaches in mice can, in principle, also be applied in other animal species. Particularly, zebra fish represent a suitable modeling organism with largely unlocked potential in pediatric solid cancer research. While zebra fish models might not be as translatable as mouse models due to their non-mammal nature, shorter generation times, higher scalability, lower costs, extracorporeal embryonic development, and skin transparency (allowing live cell imaging) render them a powerful and complementary modeling tool for pediatric tumors [150]. Good examples of such a genetically-engineered zebrafish models for pediatric sarcoma can be found in the study of Parant et al. on MPNST [151] as well as the recent review of Casey et al. on sarcoma zebrafish models for pediatric cancers [152]. For rhabdomyosarcoma an eRMS model expressing *KRAS^G12D^* in muscle satellites cells via *Rag2* promoter by Langenau et al. [153,154] as well as an aRMS model, systemically expressing the *PAX3-FOXO1* fusion by Kendall et al. [155] were developed to date. These models have already proven to be a valuable tool to deepen the understanding of RMS tumorigenesis [156]. Galindo et al. further showed that expression of *PAX3-FOXO1* in syncytial muscle fibers of Drosophila can drive sarcomagenesis and is further supported by constitutive RAS expression [157]. Apart from GEMMs, zebra fish can also serve as host organisms for engraftment models, such as PDX. While this was previously hampered by the typical 32° living conditions of zebra fish and time-restricted to the first four weeks of life when the fish’s immune system is not yet fully developed, the recently reported *prkdc*^−^*/^−^, il2rga^−^/^−^* line represents a 37°-adapted immuno-compromised zebra fish line, allowing for engraftment of several cancer types, including eRMS and drug-response monitoring via live cell imaging [158].

Canine models have also been important for sarcoma research especially for osteosarcoma. Spontaneous OS is quite common in large dogs and highly resembles human OS in terms of gene expression profiles and histological analysis [159,160]. From a genetic perspective, OS in dogs is also characterized by complex karyotypes with variable structural and numerical chromosomal aberrations and involves many of the genes important for human OS pathogenesis including *TP53*, *RB*, and *PTEN* [161,162,163]. Because osteosarcoma naturally occurs with high frequency in dogs and shares many biological and clinical similarities with osteosarcoma in humans, canine OS models have provided means to understand the disease at different levels. Most importantly, they provided an opportunity to evaluate new treatment options, and indeed the development of treatment strategies in dogs and humans has mutually benefited both species [164]. Although canines have been instrumental for OS research, it is worth noting that OS in dogs occurs exclusively in old age, not entirely mimicking the human disease that peaks in adolescence [107].

## 4. Applications of Pediatric Sarcoma Mouse Models

Most and foremost, model generation is no end in itself. Both biological and translational advances require purposeful utilization of the right model system for the respective research question. While CDXs are still the by far most commonly used model due to broad availability and ease of use, either PDX- or GEM-models are typically the most suitable model for both basic and translational research questions (Figure 5). In general, GEM models are ideal to deepen our understanding of basic sarcoma biology while PDX models are of particularly value for preclinical testing, adequately representing patient heterogeneity. While many cell-based immunotherapies can also be tested in conventional PDX models, immunotherapies requiring endogenous immune cells can either be tested in GEMMs or humanized PDX models, the latter being very costly and technically challenging thus largely limiting their use [165,166] (Figure 6b). Both, PDX and GEM models are suitable for local therapy advancement and imaging studies, a rather underrepresented field of research, given the importance of complete resection for clinical outcome [68,167].

GEMMs of different genetic makeup can also be used to assess the fraction of tumor cells with tumorigenic potential, following the notion that some tumors rely on a small fraction of cells to drive overall cell renewal and tumor growth [168]. Following this cancer stem cell idea, Buchstaller et al., for example, compared the tumorigenic potential of two engrafted MPNST GEMMs and found that transplanted MPNST cells from *Nf1*^+/−^
*Ink4a/Arf*^−/−^ tumors encompassed a 10-fold higher fraction of cells with cancer-initiating potential than MPNST cells from *Nf1*^+/−^ and *Tp53*^+/−^ tumors [169].

Given these advantages of PDXs and GEMMs, CDXs possess one natural prime advantage GEMMs and PDXs are often lacking: they entail a corresponding in vitro system, allowing for variable functional characterization and are often very well characterized. This strength paired with the high practicality of their use makes them a highly valuable tool for present and future sarcoma research.

### Considerations for Preclinical Testing

A major consideration for model utilization is how to design meaningful and translatable preclinical therapy trials. This is particularly important for pediatric sarcoma since the rarity of individual subtypes combined with the incredibly diverse array of subsets complicates rational clinical trial design. To this end, Langenau et al. provided a comprehensive and contemporary review, highlighting 10 key points to consider when designing preclinical treatment trials [170]. The key concept is to apply the same principles and guidelines used in clinical phase I, II, and III trials to the preclinical setting in a similar stepwise approach by conducting preclinical phase I, II and III trials alike [170]. This systematic approach is equally feasible for combinatorial agent testing in vivo and elucidated that some synergistic effects can be mediated by the in vivo environment and are not picked up in vitro [171]. A prerequisite of sublime importance for successful in vivo trials is the appropriate selection of promising treatment agents, based on comprehensive molecular and drug-screening in vitro data [49]. Equally important is the selection of a set of appropriate and well characterized model systems, adequately representing patient heterogeneity, including relevant patient subsets based on molecular characteristics serving as biomarkers, possibly informing about treatment response [172,173]. Connecting molecular model characterization and drug response data is an essential avenue in moving towards precision oncology in pediatrics [172]. Recent reports applying this concept by conducting single-mouse-design studies highlight the feasibility and translational relevance of this approach [33,174,175]. Approaches to use PDX models as avatars for individual patients that are parallelly being treated in the clinic are possible in principle, but typically hampered by time-consuming model establishment and variable engraftment rates [176]. Nimmervoll et al. further introduced the concept of a mouse clinic, aiming to more closely resemble the multimodal clinical therapy, including chemotherapy, radiotherapy, and local resection for testing the application of new targeted treatments [177]. While this elaborate approach will likely be to too complex and resource-intensive for general use, one should carefully consider the combination of new targeted treatments and immunotherapies with mainstays of current therapy, including local resection and radiation. A more feasible approach to deepen the insight of therapy trials is the use of molecular barcoding of engrafted cells to reveal therapy-induced clonal selection processes [178,179,180] (Figure 6c). Useful examples on how to present preclinical therapy response data can be found in the review from Gengenbacher et al. [13].

## 5. Future Directions

Ideally, the toolbox of preclinical sarcoma models should sufficiently represent the immense intertumoral heterogeneity of different sarcoma subtypes. The major limitation towards establishing enough of such highly predictive sarcoma models remains the scarcity of available patient material for research of these rare diseases. As recently outlined by Painter et al. in “The Angiosarcoma Project” as a part of the “Count me in” initiative by the Broad Institute, a more patient-centered research approach, empowering patients to directly share their experience, samples and data proved to be very successful in overcoming the logistic difficulties of non-centralized treatment of rare cancers and could potentially serve as a blueprint for pediatric solid tumors to help affected children and their families to engage in meaningful research to better future therapies [181]. More information on the patient-centered “Count me in” initiative, which has recently expanded to OS, can be found here: https://joincountmein.org/, last accessed 21 April 2021.

For existing models, a key challenge will be to make the scarcely used, but particularly representative and predictive GEMM- and PDX-models more available to both academic and industrial research in order to increase relevance and translational validity of obtained results [13]. This is often complicated by the legal frameworks accompanying the exchange of models, particularly for PDXs, since they entail genomic patient information. Increased model availability would make a broadly accepted standard for model description even more important (e.g., PDX-MI/PDX models minimal information standard) [52].

For GEMMs, more efforts should be taken towards establishing syngeneic transplantation models, allowing biobanking such as in PDX models, but with immunocompetent background. Especially immunotherapeutic approaches for children with sarcoma would immensely benefit from this approach. Importantly, once a GEMM-derived syngeneic transplantation model is established, GEM-tissue becomes expandable and the thus now scalable models can be made available beyond the host institution while costs drop from high to moderate [12] (Figure 6a). Ideally, syngeneic engraftment models (SAMs) should be accompanied by matched FCS-free 3D spheroid-/tumoroid-cultures for in vitro screening and functional studies [182,183]. This is particularly important since vast potential for future treatment advances might lie in rational combination therapies, which even more so require high-throughput in vitro screening before validation in vivo [45,61]. An additional largely unexplored potential lies in revisiting previously developed GEMMs among human samples, with up-to-date molecular profiling techniques, as exemplified by Mito et al. [184]. To this end, the recently released 285k methylation array for broad classification or the single-cell methylation analysis for mouse samples might help to deepen the knowledge on cellular origins and epigenetic determinants of various sarcoma subtypes [110,185]. To make best use of existing models, open-source platforms to access generated molecular data and microarrays to simultaneously stain for specific targets among many existing models, can be of great value to select appropriate models for particular studies [49]. Figure 6 highlights a selection of different modeling approaches that could be of additional or complementary value to various established models. For, so far, elusive GEMMs, such as EwS, which seem to require a human-specific molecular background, human–mouse chimera models could provide a solution (Figure 6d). For instance, Cohen et al. successfully developed such a model for neuroblastoma (NB) recently, by introducing NB-specific alterations on a doxycycline(Dox)-inducible vector into human stem cells (in this case hNCCs/human stem-cell derived neural crest cells) before injecting them into early mouse embryos [186]. NB-specific gene expression could later be regulated by dox administration in chimeric offspring. Another, already commonly used approach to regulate gene expression in vivo via alimental Dox are engraftment models transduced with vectors carrying a Dox-inducible RNA interference (e.g., TRE-shRNA) or inducible nuclease (e.g., Cas9-sgRNA) cassette to inducible knockdown or knockout a gene of interest [187] (Figure 6e). Such approaches are also feasible on a systemic level for to investigate the systemic toxicities due to genetic targeting of a specific gene [188,189] (Figure 6f). Genetic dropout screens via RNAi or CRISPR are also no longer limited to the in vitro setting, but can also be carried out in vivo [190].

This is also possible in models of metastasis and progression [191] (Figure 6g), a phenomenon of utmost importance for sarcoma patients, but understudied via mouse modeling [62]. Many mouse models do not metastasize spontaneously before mice have to be sacrificed due to primary tumor burden [192,193]. Thus, the most translationally relevant modeling approach for metastasis modeling is local tumor resection, typically via limb resection, and holds much promise to further elucidate mechanisms of metastasis for the various pediatric sarcoma entities, as well as possible therapeutic vulnerabilities [194,195]. At the same time, it enables utilization of PDX- and GEM models for improvement of local resection, e.g., via molecular imaging or photodynamic therapy, which, despite promising ongoing efforts, is still a rather unexplored field of pediatric sarcoma research with much unexplored potential to improve clinical outcomes [195,196,197,198,199,200]. 

## 6. Conclusions

Mouse models for pediatric sarcoma are an indispensable tool to deepen our understanding of this incredibly diverse group of diseases. Among them, genetically-engineered and patient-derived xenograft models are the most representative and predictive model types for meaningful basic and translationally relevant research. Recent technological advances, such as somatic GEM modeling, inducible in vivo gene regulation, cellular barcoding of engraftment models and first and foremost strong collaborative and international efforts to establish representative model repositories have the power to adequately represent at least major, high risk entities of the diverse landscape of pediatric sarcoma. If utilized correctly for preclinical testing, these models have the potential to transform the future clinical treatment of childhood and adolescence sarcoma.

## Figures and Tables

**Figure 1 jcm-10-01578-f001:**
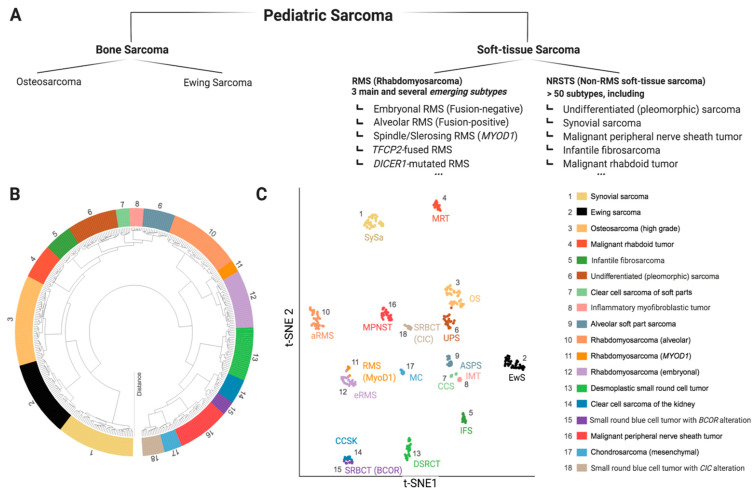
The diverse landscape of pediatric sarcoma. (**A**) Extraction from the current WHO classification of soft-tissue and bone tumors (5th edition, 2020) with focus on pediatric sarcoma [8]. (**B**,**C**) Molecular classification by whole genome DNA methylation data, here depicted for 18 sarcoma entities relevant for childhood and adolescence. Data shown as hierarchical clustering analysis, (**B**) and t-distributed stochastic neighbor embedding (t-SNE), (**C**) data adapted from Kölsche et al., 2021 [10].

**Figure 2 jcm-10-01578-f002:**
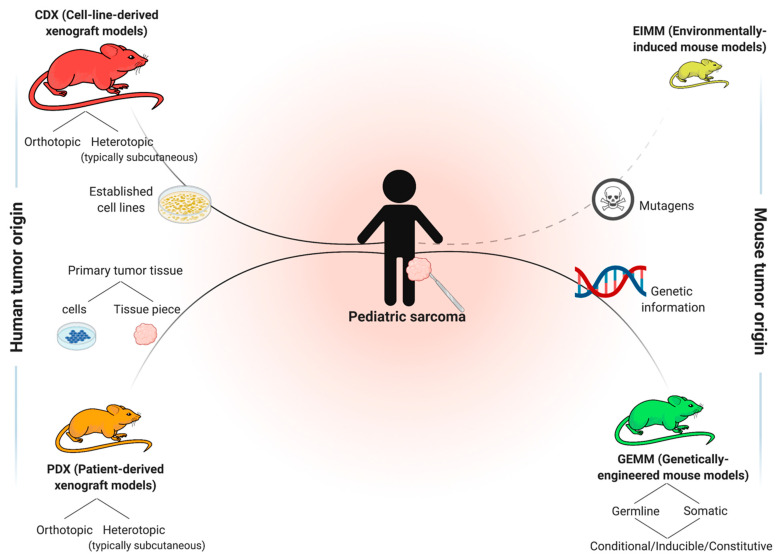
Different approaches to model sarcoma in vivo. Relative sizes of mouse pictograms resemble approximate utilization of modeling approaches in current sarcoma research. Dashed line illustrates that environmentally-induced models (EIMMs) are typically not particularly relevant for childhood sarcoma.

**Figure 3 jcm-10-01578-f003:**
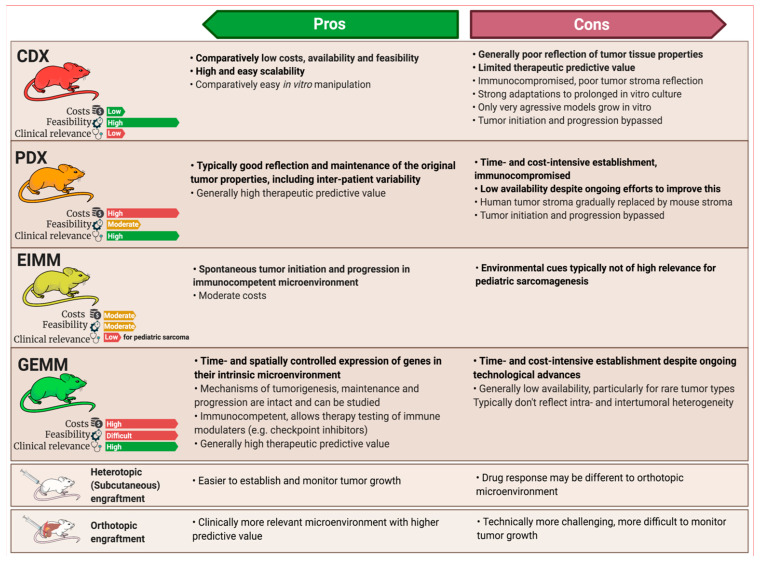
Pros and cons of different in vivo modeling approaches. While cell-line-derived xenograft models (CDXs) and patient-derived xenograft models (PDXs) are both engraftment models, environmentally-induced models (EIMMs) and genetically-engineered mouse models (GEMMs) can be utilized to establish syngeneic engraftment models (SAMs), enabling scalability for these models, too. Pros and cons of different engraftment sites depicted at the bottom apply to all of the engraftment models, regardless of origin.

**Figure 4 jcm-10-01578-f004:**
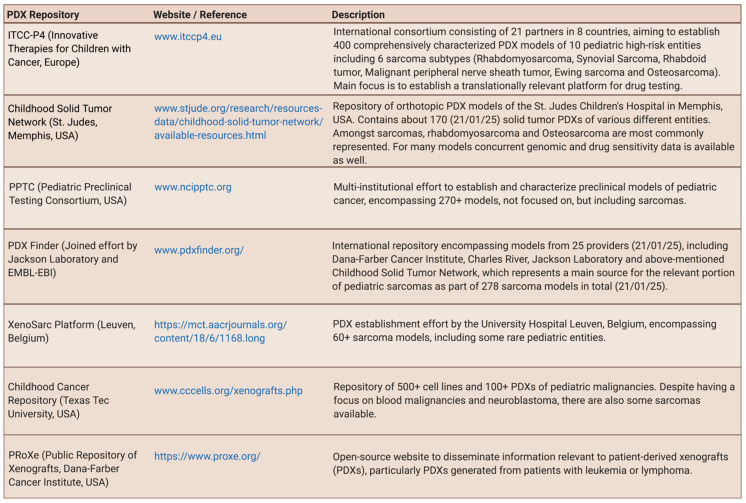
Selection of PDX-repositories relevant for pediatric sarcoma.

**Figure 5 jcm-10-01578-f005:**
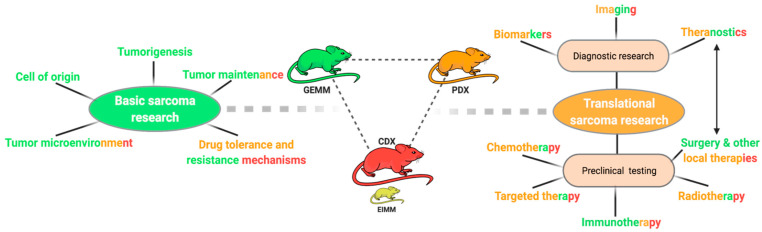
Utilization of sarcoma mouse models. Schematic overview of different research questions in sarcoma research. Color codes represent the generally most suitable modeling approach for the respective research field ((green) = GEMM, (orange) = PDX, (red) = CDX).

**Figure 6 jcm-10-01578-f006:**
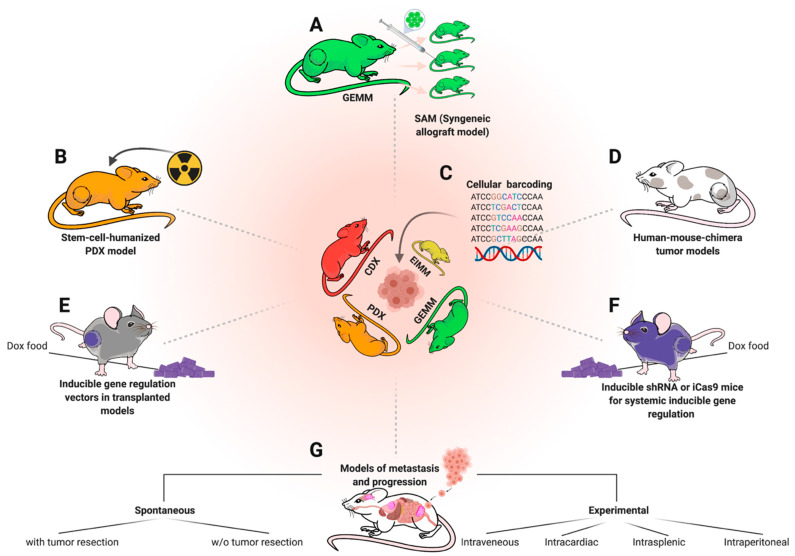
Complementary in vivo modeling approaches. The depicted methods are not specific to sarcoma, but can be applied to complement and optimize utilization of existing models and approaches. (**A**) Derivation of syngeneic engraftment models (SAMs) from GEMMs to increase scalability. (**B**) Humanization of existing PDX models for immunotherapy trials requiring an endogenous immune system, e.g., checkpoint inhibitor therapies. (**C**) Cellular barcoding of engraftment models to study clonal selection effects under therapy and metastasis. (**D**) Human mouse chimera genetic models to bypass mouse-human biology discrepancies. (**E**) Transduction of tumor cells from engraftment models using Doxycycline (Dox)-inducible vectors (e.g., TRE-shRNA to inducible knockdown a gene of interest) to investigate molecular dependencies in vivo. (**F**) Applying the same approach (**E**) on a systemic level to additionally study systemic toxicity in a target-gene-dependent fashion. (**G**) Overview of divergent metastasis modeling approaches.

## Data Availability

No new data were created in this study. Data sharing is not applicable to this article.

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
