# Peer review of "Preclinical In Vivo Modeling of Pediatric Sarcoma—Promises and Limitations"

_jcm, 2021, doi:10.3390/jcm10081578_

Round 1

Reviewer 1 Report

As a reviewer, I believe the article review structured very well and is inclusive on the title that chosen by authors as review title!

Author Response

We thank the reviewer for taking the time to read our manuscript and for the positive feedback.

Reviewer 2 Report

This is a very comprehensive review of the currently available mouse models in pediatric sarcoma research. It starts by reviewing the recognized soft tissue sarcoma subtypes based on methylation profile analysis and the describes the different mouse model approaches. This section is a rather generic discussion of the different pros and cons, and while very nicely illustrated, it could probably be shortened a bit. The following section about the different repositories of available sarcoma PDX models then is very useful resource for the field, as is the summary of the mouse eRMS and aRMS models. The sections about the very rare soft tissue sarcoma subtypes without available mouse models could probably also be shortened and summarized a bit. Finally, the application of the models to translational studies I found again very useful. In summary, this is a carefully written review of the current models available for sarcoma research and clearly warrants publication in JCM.

Author Response

We thank the reviewer for taking the time to read our manuscript and for the positive feedback. We agree that some sections can be shortened. We have revised the text to implement these changes. Particularly section 3.4 was shortened and all rare subtypes to which there are no current models to date were grouped into one subsection. An updated version word file showing track changes is provided.

Reviewer 3 Report

This review does a comprehensive job in summarizing the current state of in vivo experimental models for pediatric sarcoma. It further highlights the main issues and active areas of discussion regarding in vivo modeling for pediatric cancer in general including the pros & cons of the various in vivo models, the role and potential applications of in vivo models, the various limitations of each model type, and the need for a collaborative and harmonized effort to better integrate in vivo modeling to clinical care and enable future research efforts. 

My comments and criticisms are as follows:

1) While the Kolsche et al. 2021 referenced material (reference #10) uses the term "small blue round cell tumor" (e.g in Figure 1), note that general usage has been "small round blue cell tumor". In fact, the authors utilize this histologic label in section 3.4.15. I recommend review of the manuscript text for consistency.

2) In Figure 2 which graphically represents different approaches to sarcoma modeling, the authors differentiate between "orthotopic" vs "subcutaneous" implantations in CDX and PDX models. Perhaps the more appropriate dichotomy is "orthotopic" vs "heterotopic" in which the subcutaneous route representing the most common type of heterotopic modeling.

3) In Table 1, please note that there is a typographical error under the "Subcutaneous engraftment" section. The word "microenvironment" is misspelled. Also, consider the suggestion made in comment #2 above regarding replacement of "subcutaneous" with "heterotopic". Lastly, the Table 1 legend states "While CDXs and PDXs are always engrafted..." which I would contend is misleading. While I agree that injected cells from established lines used to generate CDXs generally show ~100% engraftment (though it is clear that not all established lines are amenable to CDX generation), it is not a true statement that PDXs "always engraft". In fact, engraftment in PDXs can range from 30 - 60% depending on the reference (average engraftment is ~30-40%) and can also vary within a given model with successive generations. Please clarify this statement.

4) Under section 3.2 in which the authors describe existing PDX repositories, I would also highlight the fact that in addition to underrepresentation of pediatric-specific PDXs (particularly for ped sarcoma), some repositories (e.g. the referenced https://repositive.io) do not even allow for filtering by age further highlighting that existing repositories not only underrepresent pediatric models, but are also not necessarily poised to encourage deposition of pediatric models. The authors do a great job highlighting these inequities.

5) Table 2 summarizes existing repositories. I would also include the PROXe repository (Dr David Weinstock at Dana Farber Cancer Institute) which also houses pediatric PDX models (proxe.org). Memorial Sloan Kettering Cancer Center's Department of Pediatrics will also soon be releasing their collection of pediatric PDX models (>250 PDX models representing >160 discrete cases with a high proportion of sarcoma histologies).

6) Line 220, typographical error. Currently "MyOD-positive progenitors", and likely should say "MyoD-positive progenitors..."

7) Line 280, typographical error. Currently "Van Mater et applied...", and should say "Van Mater et al. applied...."

8) Line 311, typographical error. Currently "no eRMS9", and should say "no eRMS)".

9) The authors switch between use of "Trp53" and "TP53" throughout the manuscript. I would review the manuscript and use consistent and appropriate usage. For example, line 314 uses "Trp53" and line 374 uses "TP53". There are numerous other examples throughout the paper. Though this is a minor point, it may be confusing or distracting to readers.

10) For the Ewing sarcoma GEMM section 3.4.6, the authors describe the canonical EWSR1-FLI1 fusion that characterizes Ewing sarcoma. I suggest also mentioning that the diagnosis of Ewing sarcoma is further characterized by EWS-ets fusions since alternative EWS fusion partners exist, and that these are to be distinguished from the "Ewing-like sarcomas" which the authors later refer to under 3.4.14 (BCOR altered sarcomas) and 3.4.15 (CIC altered sarcomas). 

11) For section 3.4.7 Synovial sarcoma, I suggest removing the "monomorphic" (line 419) term as a general description of synovial sarcoma since synovial sarcoma can manifest as either a monormorphic subtype or biphasic subtype which the authors later mention (line 428).

12) For section 3.4.9 Infantile fibrosarcoma, a variant infantile fibrosarcoma model has previously been generated albeit with a non-canonical fusion partner (EML4-NTRK3 vs the more common ETV6-NTRK3 fusion). See https://www.ncbi.nlm.nih.gov/pmc/articles/PMC4850889/.

13) For sections 3.4.11 and 3.4.12, the abbreviation used is incorrect and inconsistent. Please correct to consistently say "CCSK" (and not CSSK) and "CCS" (and not CSS).

14) Similar to comment #13 above, for sections 3.4.14 and 3.4.15, the abbreviation should be corrected to "SRBCT" and not "SBRCT".

15) I would reference the statement regarding SRBCT-CIC as a distinct subtype. For example, https://www.ncbi.nlm.nih.gov/pmc/articles/PMC4108073/.

16) The authors state that no GEM models to date have been developed for BCOR-altered and CIC-altered SRBCTs, as well as for DSRCT and mesenchymal chondrosarcoma. Perhaps this could be consolidated into one section particularly since the GEM modeling section is already quite expansive in breadth. 

17) In section 3.5, the authors describe alternative in vivo models, particularly use of zebrafish models. The authors reference the work of Dr David Langenau and Dr Jim Amatruda who have performed intriguing work using these models to study both the biology but also to serve as in vivo drug screening models that are very much translationally relevant. As such, I disagree with the author's statement or implication that these models "are not as translationally relevant" (line 546). Please elaborate on this statement.

18) For section 3.5 describing non-murine models, the final paragraph very briefly describes the utilization of canine models (lines 565 - 569). As this is an overall and comprehensive review of pediatric sarcoma models, it is important to provide readers with a broader description of the use of canine models particularly in studying Osteosarcoma. The authors only provide a very cursory comment on this topic which is quite important to describe to readership since canine models have also served as the backbone for our understanding of osteosarcoma. Furthermore, work using canine models have been instrumental in the translation of prior, current and future clinical trials in osteosarcoma.

19) In Section 4 describing the applications of pediatric sarcoma mouse models, while the authors comment that CDXs are the most commonly used in vivo models for study, I would suggest the authors comment on what settings or situations use of CDX models are appropriate. As the authors describe throughout the manuscript, use of alternative models like PDXs or GEMMs may not be readily accessible to investigators. Hence, the continued use of CDX models. The authors seem to imply, whether they meant to or not, that CDX models are not good models for in vivo studies. I contend that there are situations in which CDX models will continue to be useful. I request the authors comment on this and discuss/describe how CDXs models should be used, and, if not, why they think so.

20) Regarding Figure 3, I would again challenge the authors to review this figure since it currently suggests that CDXs have no role in experimental modeling (i.e. there are no connecting lines to either the left or right side describing the different types of work that in vivo models are used for).

21) Please note that "Future directions" is labeled "4", but should be section "5" (line 628). 

22) Under "Future directions", in addition to the Angiosarcoma Project mentioned in line 633, this program is a part of the broader Count Me In project. The authors may also want to mention the "Osteosarcoma Project" which has also been developed under the umbrella of Count Me In and championed by MIB agents. See https://osproject.org/

23) Please revise use of "complexion" in line 692. This sentence does not make sense. In general, please review the manuscript for similar misuse of terms and grammar.

24) The authors mention that metastasis modeling using "local tumor resection" (lines 695-697) is unclear. I presume the authors are referring to amputation modeling to study mechanisms of metastasis as used in osteosarcoma and rhabdomyosarcoma in vivo modeling. Please clarify.

Author Response

We thank the reviewer for the comprehensive review of our manuscript and the excellent suggestions.  We have implemented all changes suggested. An updated version word file showing track changes is provided. Please see below a point by point response.

1.While the Kolsche et al. 2021 referenced material (reference #10) uses the term "small blue round cell tumor" (e.g in Figure 1), note that general usage has been "small round blue cell tumor". In fact, the authors utilize this histologic label in section 3.4.15. I recommend review of the manuscript text for consistency.

We changed the text in Figure 1 as well as the text body to “small round blue cell tumor” for consistency as suggested.

2.In Figure 2 which graphically represents different approaches to sarcoma modeling, the authors differentiate between "orthotopic" vs "subcutaneous" implantations in CDX and PDX models. Perhaps the more appropriate dichotomy is "orthotopic" vs "heterotopic" in which the subcutaneous route representing the most common type of heterotopic modeling.

We agree. We changed the text in Figure 2 to “heterotopic (typically subcutaneous)” as suggested.

3.In Table 1, please note that there is a typographical error under the "Subcutaneous engraftment" section. The word "microenvironment" is misspelled. Also, consider the suggestion made in comment #2 above regarding replacement of "subcutaneous" with "heterotopic". Lastly, the Table 1 legend states "While CDXs and PDXs are always engrafted..." which I would contend is misleading. While I agree that injected cells from established lines used to generate CDXs generally show ~100% engraftment (though it is clear that not all established lines are amenable to CDX generation), it is not a true statement that PDXs "always engraft". In fact, engraftment in PDXs can range from 30 - 60% depending on the reference (average engraftment is ~30-40%) and can also vary within a given model with successive generations. Please clarify this statement.

The typographical error was corrected. “Subcutaneous” was changed to “Heterotopic (Subcutaneous)”. We also agree with the reviewer’s statement that not all PDXs models engraft. We rewrote the respective sentence in the legend of table 1 to clarify the intended message: “While CDXs and PDXs are both engraftment models, EIMMs and GEMMs can also be utilized to establish syngeneic engraftment models (SAMs), enabling scalability for these models, too.”

  1. Under section 3.2 in which the authors describe existing PDX repositories, I would also highlight the fact that in addition to underrepresentation of pediatric-specific PDXs (particularly for ped sarcoma), some repositories (e.g. the referenced https://repositive.io) do not even allow for filtering by age further highlighting that existing repositories not only underrepresent pediatric models, but are also not necessarily poised to encourage deposition of pediatric models. The authors do a great job highlighting these inequities.

We thank the reviewer for stressing these issues and adapted the text  to acknowledge them: “Furthermore, since many don’t allow filtering available models by age group, it appears that they are often not poised to encourage deposition of pediatric PDXs.” (lines  175-177, page 5).

5.Table 2 summarizes existing repositories. I would also include the PROXe repository (Dr David Weinstock at Dana Farber Cancer Institute) which also houses pediatric PDX models (proxe.org). Memorial Sloan Kettering Cancer Center's Department of Pediatrics will also soon be releasing their collection of pediatric PDX models (>250 PDX models representing >160 discrete cases with a high proportion of sarcoma histologies).

Thank you for pointing out these two valuable additional repositories. We included the PROXe repository in table 2 and referenced the soon-to-be-released MSKCC one in the text (Line 167-169, page 5).

6) Line 220, typographical error. Currently "MyOD-positive progenitors", and likely should say "MyoD-positive progenitors..."

7) Line 280, typographical error. Currently "Van Mater et applied...", and should say "Van Mater et al. applied...."

8) Line 311, typographical error. Currently "no eRMS9", and should say "no eRMS)".

All typographical errors mentioned above (points 6-8) have been corrected. Thanks for pointing this out.

9) The authors switch between use of "Trp53" and "TP53" throughout the manuscript. I would review the manuscript and use consistent and appropriate usage. For example, line 314 uses "Trp53" and line 374 uses "TP53". There are numerous other examples throughout the paper. Though this is a minor point, it may be confusing or distracting to readers.

We made this distinction to differentiate mouse and human p53, as sometimes we are referring to the mutations in patients and other times to genotypes in mice. We re-evaluated this issue to ensure consistent use of TP53 for human and Tp53 for the mouse gene (and not Trp53 which seems to be a older alternative gene symbol for the mouse gene).

10) For the Ewing sarcoma GEMM section 3.4.6, the authors describe the canonical EWSR1-FLI1 fusion that characterizes Ewing sarcoma. I suggest also mentioning that the diagnosis of Ewing sarcoma is further characterized by EWS-ets fusions since alternative EWS fusion partners exist, and that these are to be distinguished from the "Ewing-like sarcomas" which the authors later refer to under 3.4.14 (BCOR altered sarcomas) and 3.4.15 (CIC altered sarcomas). 

We agree. We have now edited the text to reflect this:  “EWSR1-ETS (typically FLI1)” (line 428, page 10).

11) For section 3.4.7 Synovial sarcoma, I suggest removing the "monomorphic" (line 419) term as a general description of synovial sarcoma since synovial sarcoma can manifest as either a monormorphic subtype or biphasic subtype which the authors later mention (line 428).

We agree and removed the term monomorphic as suggested.

12) For section 3.4.9 Infantile fibrosarcoma, a variant infantile fibrosarcoma model has previously been generated albeit with a non-canonical fusion partner (EML4-NTRK3 vs the more common ETV6-NTRK3 fusion). See https://www.ncbi.nlm.nih.gov/pmc/articles/PMC4850889/.

We thank the reviewer for pointing this out. The transformative potential in NIH3T3 has also been described for the canonical ETV6-NTRK3 fusion before including engraftment into immunocompromised mice (PMID: 10702799). We therefore added both of these references to the text body (lines 510-514, page 12) and supplementary table 1.

13) For sections 3.4.11 and 3.4.12, the abbreviation used is incorrect and inconsistent. Please correct to consistently say "CCSK" (and not CSSK) and "CCS" (and not CSS).

We have corrected spelling to CCSK and CCS as pointed out by the reviewer.

14) Similar to comment #13 above, for sections 3.4.14 and 3.4.15, the abbreviation should be corrected to "SRBCT" and not "SBRCT".

We changed to SRBCT for consistency as suggested.

15) I would reference the statement regarding SRBCT-CIC as a distinct subtype. For example, https://www.ncbi.nlm.nih.gov/pmc/articles/PMC4108073/.

Thanks for pointing this out. The reference was added.

16) The authors state that no GEM models to date have been developed for BCOR-altered and CIC-altered SRBCTs, as well as for DSRCT and mesenchymal chondrosarcoma. Perhaps this could be consolidated into one section particularly since the GEM modeling section is already quite expansive in breadth. 

We agree and grouped all rare subtypes to which there are no current models to date into one subsection. We also shortened wherever possible.

17) In section 3.5, the authors describe alternative in vivo models, particularly use of zebrafish models. The authors reference the work of Dr David Langenau and Dr Jim Amatruda who have performed intriguing work using these models to study both the biology but also to serve as in vivo drug screening models that are very much translationally relevant. As such, I disagree with the author's statement or implication that these models "are not as translationally relevant" (line 546). Please elaborate on this statement.

We acknowledge this valuable point of criticism and rephrased the sentence to point out the phylogenetic differences and pros and cons of mouse vs. fish (Line 614-617, page 13): “While zebra fish models might not be as translatable as mouse models due to their non-mammal nature, shorter generation times, higher scalability, lower costs, extracorporeal embryonic development and skin transparency (allowing live cell imaging) render them a powerful and complementary modeling tool for pediatric tumors [154].”

18) For section 3.5 describing non-murine models, the final paragraph very briefly describes the utilization of canine models (lines 565 - 569). As this is an overall and comprehensive review of pediatric sarcoma models, it is important to provide readers with a broader description of the use of canine models particularly in studying Osteosarcoma. The authors only provide a very cursory comment on this topic which is quite important to describe to readership since canine models have also served as the backbone for our understanding of osteosarcoma. Furthermore, work using canine models have been instrumental in the translation of prior, current and future clinical trials in osteosarcoma.

We agree and have edited the text to further elaborate on the value of canine osterosarcoma models (lines 680-693, page 14): “Canine models have also been important for sarcoma research especially for osteosarcoma. Spontaneous OS is quite common in large dogs and highly resembles human OS in terms of gene expression profiles and histological analysis [162,163]. From a genetic perspective, OS in dogs is also characterized by complex karyotypes with variable structural and numerical chromosomal aberrations and involves many of the genes important for human OS pathogenesis including TP53, RB, and PTEN [164–166]. Because osteosarcoma naturally occurs with high frequency in dogs and shares many biological and clinical similarities with osteosarcoma in humans, canine OS models have provided means to understand the disease at different levels. Most importantly, they provided an opportunity to evaluate new treatment options, and indeed the development of treatment strategies in dogs and humans has mutually benefited both species [167]. Although canines have been instrumental for OS research, it is worth noting that OS in dogs occurs exclusively in old age, not entirely mimicking the human disease that peaks in adolescence[109].

19) In Section 4 describing the applications of pediatric sarcoma mouse models, while the authors comment that CDXs are the most commonly used in vivo models for study, I would suggest the authors comment on what settings or situations use of CDX models are appropriate. As the authors describe throughout the manuscript, use of alternative models like PDXs or GEMMs may not be readily accessible to investigators. Hence, the continued use of CDX models. The authors seem to imply, whether they meant to or not, that CDX models are not good models for in vivo studies. I contend that there are situations in which CDX models will continue to be useful. I request the authors comment on this and discuss/describe how CDXs models should be used, and, if not, why they think so.

We agree. CDX model are and will continue to be very useful models for sarcoma research. We therefore added a short paragraph to section 4 to clarify this: “Given these advantages of PDXs and GEMMs, CDXs possess one natural prime advantage GEMMs and PDXs are often lacking: They entail a corresponding in vitro system, allowing for variable functional characterization and are often very well characterized. This strength paired with the high practicality of their use makes them a highly valuable tool for present and future sarcoma research.” (Lines 712-717, page 14).

20) Regarding Figure 3, I would again challenge the authors to review this figure since it currently suggests that CDXs have no role in experimental modeling (i.e. there are no connecting lines to either the left or right side describing the different types of work that in vivo models are used for).

We thank the reviewer for this point and agree as elaborated in point 19. We revisited figure 3 and adapted the color code to clarify that CDXs still have broad applicability.

21) Please note that "Future directions" is labeled "4", but should be section "5" (line 628). 

We corrected this typographical error. We also changed the “Conclusions” section from “5” to “6” accordingly.

22) Under "Future directions", in addition to the Angiosarcoma Project mentioned in line 633, this program is a part of the broader Count Me In project. The authors may also want to mention the "Osteosarcoma Project" which has also been developed under the umbrella of Count Me In and championed by MIB agents. See https://osproject.org/

Thanks for this suggestion, which we gladly followed by adapting the first paragraph of section 5 to include the ‘Osteosarcoma Project’ as well. (Line 786-788, page 16).

23) Please revise use of "complexion" in line 692. This sentence does not make sense. In general, please review the manuscript for similar misuse of terms and grammar.

We have revised the text for misuse of grammar and rephrased the sentence in line 692. (Now line 842).

24) The authors mention that metastasis modeling using "local tumor resection" (lines 695-697) is unclear. I presume the authors are referring to amputation modeling to study mechanisms of metastasis as used in osteosarcoma and rhabdomyosarcoma in vivo modeling. Please clarify.

We rephrased this sentence to increase clarity here: “Thus, the most translationally relevant modeling approach for metastasis modeling is local tumor resection, typically via limb resection, and holds much promise to further elucidate mechanisms of metastasis for the various pediatric sarcoma entities as well as possible therapeutic vulnerabilities[195,196].”  (Lines 846-, page 18)